# The Effects of Dietary Inclusion of Mulberry Leaf Powder on Growth Performance, Carcass Traits and Meat Quality of Tibetan Pigs

**DOI:** 10.3390/ani12202743

**Published:** 2022-10-12

**Authors:** Sutian Wang, Cuiming Tang, Jianhao Li, Zhenjiang Wang, Fanming Meng, Guoqing Luo, Haiyun Xin, Jianwu Zhong, Yuan Wang, Baohong Li, Zhiyi Li, Lian Chen, Bin Hu, Sen Lin

**Affiliations:** 1Key Laboratory of Livestock and Poultry Breeding, Guangdong Key Laboratory of Animal Breeding and Nutrition, Institute of Animal Science, Guangdong Academy of Agricultural Sciences, Guangzhou 510640, China; 2Laboratory of Urban Agriculture in South China, Sericultural & Agri-Food Research Institute, Guangdong Academy of Agricultural Sciences, Guangzhou 510640, China; 3Maoming Branch, Guangdong Laboratory for Lingnan Modern Agriculture, Maoming 525000, China

**Keywords:** Tibetan pigs, mulberry leaf powder, meat quality, muscle fiber, lean percentage, antioxidant activity

## Abstract

**Simple Summary:**

In recent years, consumers have been paying more and more attention to the flavor and texture of pork, and, therefore, the Chinese indigenous pig breeds, like Tibetan pigs, are valued for good meat quality. Herein, we investigated whether dietary mulberry leaf powder could further improve the meat quality of Tibetan pigs. Our results indicated improved meat quality in Tibetan pigs who consumed mulberry leaf powder, which might be the consequence of enhanced antioxidant activity and altered muscle fiber type and morphology.

**Abstract:**

This research was conducted to study the effects of dietary inclusion of mulberry leaf powder (MLP) on growth performance, meat quality, antioxidant activity, and carcass traits of Tibetan pigs. Eighteen Tibetan pigs (33.8 ± 1.1 kg) were assigned to two treatment groups randomly and received either the control diet (CON) or a basal diet containing 8% MLP (MLP) for two months. After the two-month feeding trial, the MLP group showed lower backfat thickness while a higher lean percentage. Compared with CON pigs, MLP pigs had higher serum CAT activity. In addition, dietary MLP supplementation significantly decreased the muscle shear force. Muscle fiber morphology analysis showed that MLP pigs had larger muscle fiber density while smaller muscle fiber cross-sectional area. Up-regulated gene expression of myosin heavy chain (MyHC)IIa was also observed in MLP pigs. These results indicate that the enhanced antioxidant activity and altered muscle fiber type and morphology appeared to contribute to the improvement of meat quality in Tibetan pigs fed diets containing MLP.

## 1. Introduction

With special sensory attributes like tender texture, richness in nutrition composition, and unique flavor, pork is popular with customers worldwide [1,2]. As is reported by the Food and Agriculture Organization of the USA, the production and consumption of pork worldwide account for at least 30% of the total meat market. Of note, China is the largest market for pork consumption and production, accounting for approximately 48.7% of total production in the whole world [3,4].

In China, the domestication of pigs can trace back to as early as 9000 years ago [5]. Traditionally, most households in China raise a few pigs and slaughter them for important ceremonies such as weddings and the Spring Festival [6]. In China, Pig breeding by countless farmers over millennia led to the creation of a wide range of indigenous pig breeds [6], and pork from indigenous pigs served as an important ingredient for delicious Chinese food. Over the past decades, with the industrialization of pig farming, an increasing amount of pork in China has been supplied by large-scale livestock companies, where most pigs raised are industrial hog breeds (i.e., Yorkshire).

In the last few years, consumers’ individual needs are growing rapidly. Meanwhile, Chinese indigenous pork is becoming increasingly popular among consumers, of which Tibetan pork is the most popular one because of its unique edible quality [7]. As a peculiar pig breed of China, the Tibetan pig mainly lives in the Qinghai–Tibet plateau [8] and has evolved special living habits and genetic characteristics. Thousands of years living in the Qinghai–Tibet plateau renders Tibetan pigs the ability to live in hypoxia and a high-frigid environment. The harsh living habitat forces Tibetan pigs to develop extremely strong adaptability and stress resistance. In the Qinghai–Tibet plateau, Tibetan pigs live all year round under poor feeding and management conditions. The feeding mode of Tibetan pigs is quite primitive and close to the way how organic food is produced, and, therefore, Tibetan pork is considered to be eco-friendly and safe. In addition, Tibetan pork is more healthy because it contains more essential amino acids and unsaturated fatty acids [9].

Compared with commercial pig breeds, the feeding periods of Tibetan pigs are much longer, which last 12–18 months. Therefore, more fat is deposited, and high content of subcutaneous fat is characterized in Tibetan pigs [10]. Although Tibetan pork is becoming increasingly popular, too much subcutaneous fat appears to be a disadvantageous factor affecting the meat quality and choice of new customers. Moreover, high-fat ingestion can threaten the health of consumers. Therefore, strategies decreasing subcutaneous fat while keeping the meat quality of Tibetan pigs are needed to increase the market for Tibetan pork and protect the health of consumers.

Mulberry leaf, a traditional Chinese herbal medicine, has been proven to have various physiological functions, including anti-bacterial, anti-inflammatory, and antioxidative properties [11]. In recent years, mulberry leaf powder (MLP) and bioactive constituents extracted from mulberry leaves have been used in animal feeds and appeared to have beneficial effects on nutrient digestion, anti-oxidative capacity, immune function, and especially meat quality [12,13,14,15,16]. Additionally, bioactive constituents from mulberry could repress fat deposition probably through elevating leptin-stimulated lipolysis [2,17,18].

It is noteworthy that the application of MLP in pigs has mainly been focused on industrial hog breeds (i.e., Duroc × Landrace × Yorkshire) [15,19,20]. Given the different growth rates and fat deposition patterns of Tibetan pigs, it is necessary to evaluate the application effects of MLP in Tibetan pigs. Therefore, we hypothesized that dietary inclusion of MLP could decrease subcutaneous fat while improving lean percentage and meat quality in Tibetan pigs.

## 2. Materials and Methods

The protocols and procedures for animal care and handling were approved by the Animal Care and Use Committee of the Guangdong Academy of Agricultural Sciences (authorization number GAASIAS-2022006).

### 2.1. MLP Preparation

Mulberry leaves used in this study were collected from the National germplasm garden of mulberry—South China distribution in Guangdong Academy of Agricultural Sciences. When collected, mulberry leaves were dried using a dryer, powdered using a feed grinder, and passed 90 mesh sieves. The nutrient composition of MLP was assayed (Table 1).

### 2.2. Animals, Experimental Design, and Diets

A total of eighteen 6-month-old Tibetan pigs (half male, half female) with close initial body weight (33.8 ± 1.1) were randomly selected from a population in our Tibetan pig breeding farm and randomly assigned to two dietary groups with nine pigs in each treatment group. In this study, only 9 pigs were included in each group because the Tibetan pig is a relatively rare pig breed and has low productivity. The litter size of Tibetan pigs can be as small as 5~6, which is much smaller than commercial pig breeds. Pigs received a basal diet were designated as control (CON) pigs, whereas pigs that received a basal diet supplemented with 8% MLP were designated as MLP pigs. The dietary formula in this experiment was based on and adapted from previous research [21] and designed to meet or exceed the nutrient requirement for Tibetan pigs (Table 2). The MLP-supplemented diet was prepared by mixing MLP powders evenly with the basal diet. The feeding trial lasted for 60 days. Pigs in each group were housed individually in environmentally controlled confinement pens that had partial concrete slatted floors. During the trial period, all pigs were allowed to have free access to water and feed (in loose form). The feed intake of pigs was recorded daily, and the average daily feed intake (ADFI) was calculated. The initial Body weight (BW), BW on the 30th day of trial, and final BW of each pig were recorded, and the average daily gain (ADG) of each pig was subsequently calculated. The feed intake and BW data were used to calculate the feed-to-gain ratios of the pigs.

### 2.3. Sample Collection

On the last day of the study, following overnight fasting, blood was drawn from the jugular veins of all 9 pigs in each group (N = 9) and set for 30 min. Serum samples were then separated by centrifugation at 3000× *g*, 4 °C for 10 min, and stored at −20 °C till analysis. Before carcass trait measurement and muscle sample collection, 6 pigs from each treatment group were randomly selected (N = 6) and transported by transport cage to the slaughterhouse in the Institute of animal science, Guangdong Academy of Agricultural Sciences. All pigs were slaughtered by electric shocks under the guidance of the working staff of the slaughterhouse. To determine the meat quality indicators (meat color, PH, dripping loss, cooking loss, shear force, and marbling score), longissimus thoracis (LT) muscle samples were obtained between the 10th to 13th rib of each pig from the left side of each carcass. The LT muscle samples collected between the 9th to 10th rib were either stored at −80 °C after snap frozen or fixed in 4% paraformaldehyde.

### 2.4. Measurement of Carcass Traits and Meat Quality

On the last day of the experiment, the backfat thickness of each pig was recorded by measuring the fat thickness at the 10th rib with a caliper. The lean percentage of each pig was calculated by using the following equation: (23.568 + (0.503 × hot carcass weight)−(21.348 × fat thickness))/HCW × 100. Meat quality characteristics were assessed by determining the following parameters of the LT muscle: color, PH, shear force, and dripping loss. 5-cm thick fillets were cut from the LT muscle sample and used to determine the meat color at 45 min and 24 h postmortem. Meat color parameters, including lightness, redness, and yellowness, were measured using a CR-400 Chroma meter (Konica Minolta Sensing Ins., Osaka, Japan) to acquire three readings for each fillet according to previous research [22]. The PH value of each LT muscle fillet at 45 min and 24 h following slaughter was determined three times using a portable pH meter (HI 9024C; HANNA Instruments, Woonsocket, RI, USA) as described previously [23]. The cooking loss and the drip loss were determined as described before [24,25]. A marbling score was given to each meat slice by comparing it with the standard scoring pictures published by the National Pork Producers Council [26]. Shear force was determined according to previous research [23]. Briefly, LT muscle samples were boiled in a 75 °C water bath to reach an internal temperature of 70 °C. After cooling down, the cooked samples were cut into 2 cm long stripes, and shear force (N) was determined by cutting slices of LT muscle samples with a digital-display-muscle tenderness meter (C-LT3B, Tenovo, Harbin, China).

### 2.5. H&E Staining and Immunofluorescence

To assess the muscle fiber characteristics, H&E staining was performed in accordance with the methods of Wang et al., 2017. Briefly, the LT muscle samples stored in 4% paraformaldehyde were dehydrated with different concentrations of ethanol. Subsequently, the samples were embedded in paraffin and cut into 4-μm thick sections with a microtome. H&E staining was then performed. H&E stained slides were observed by using the Nikon DS-U3 Image system (Nikon, Tokyo, Japan). Analysis was done using Image-pro-plus-6.0 software (Media Cybernetics, Inc., Bethesda, MD, USA). Approximately 5 representative views were selected to count the cross-sectional area of the myofiber. Myofiber density was determined using the following equation: total number of fibers counted/total area of fibers measured. 

To determine the slow-twitch and fast-twitch fibers, immunofluorescence was performed. Sections at the thickness of 10 μm were prepared by cutting paraffin-embedded LT muscle samples using a microtome. Before incubation with primary antibodies (Abcam, Cambridge, MA, USA), the sections were blocked by bovine serum albumin (3%) for 30 min overnight. The sections were then washed with PBS 3 times, and then secondary antibodies (Sevicebio, Wuhan, China) were added. DAPI was then used to restain the nucleus. The slides were observed under an inverted fluorescence microscope (Nikon Eclipse Ti-sr), and slow-twitch and fast-twitch fibers were counted by using Image-pro plus-6.0 software.

### 2.6. Analysis of Antioxidant Activities in the Serum

Activities of catalase (CAT), superoxide dismutase (SOD), and malondialdehyde (MDA) in serum collected from Tibetan pigs at the end of the study were determined by a biochemical method using biochemical reagent kits (Cat. No. A007-1-1, A00-1-1, A003-1-1, Nanjing Jiancheng Bioengineering institutes, Nanjing, China) in accordance with the manufacturer’s instructions.

### 2.7. Quantitative Real-Time PCR Analysis

LT muscle samples were ground and powdered in liquid nitrogen to extract total RNA using RNAiso Plus (Takara, Dalian, China), chloroform, isopropanol, and ethanol through centrifugation, separation, and precipitation. The collected RNA was dissolved in DEPC-treated water to prevent contamination and degradation. The quality of RNA was examined using agarose gel electrophoresis and ethidium bromide staining. Then, a reverse-transcription reaction was performed using the PrimeScript RT reagent kit (Takara, Dalian, China) in accordance with the manufacturer’s instructions, and thus, cDNA was acquired. Subsequently, the cDNA was amplified using SYBR green dye (Takara, Dalian, China) on a CFX96 Real-Time PCR System (Bio-Rad) using the following procedure: 95 °C for 30 s then 40 cycles of 95 °C for 5 s and 60 °C for 30 s. The primer sequences used are shown in Table 3. The relative expression of target genes was calculated using the 2^–ΔΔCt^ method [27].

### 2.8. Western Blot

The protocols of Western blot in this study were adapted from previous research [28]. Briefly, frozen LT muscles were ground to powder with liquid nitrogen to extract total protein using RIPA lysis buffer (Beyotime Biotech Inc, Shanghai, China). The concentrations of the protein samples were determined using the Pierce BCA protein Assay kit (Cat. No. 23227, Thermo-Fisher-Scientific, Waltham, MA, USA). Subsequently, the protein samples were loaded for SDS-PAGE. Proteins were then transferred to the PVDF membrane. β-actin protein was used as a control for equal protein loading. PVDF membrane was blocked by skimmed milk for one hour before incubation with primary antibody (anti-PGC1-α, Cat. No. 2178S, Cell Signaling Technology, Danvers, MA, USA) overnight at 4 °C. Subsequently, the PVDF membrane was incubated with secondary antibody at 4 °C for an hour. Finally, visualization was performed using the ECL kit and gel detection system (Bio-Rad, Hercules, CA, USA). Targeted bands were quantified using Image J software (NIH, Bethesda, MD, USA).

### 2.9. Statistical Analysis

All data were firstly tested for normal distribution by the Shapiro–Wilk normality test. Then, the statistical differences between the two treatment groups concerning growth performance, carcass traits, serum antioxidant parameters, meat quality, muscle fiber morphology traits, and gene and protein expression levels were all compared by using a two-tailed Student’s t-test with GraphPad Prism 8.0 software (GraphPad Prism Software Inc., San Diego, CA, USA). Differences between the two treatments were declared significant at *p* < 0.05. All data are expressed as means ± standard errors of the mean (SEM).

## 3. Results

### 3.1. Growth Performance

Results concerning growth performance and carcass traits are shown in Table 4. Compared with pigs consuming control diets, pigs consuming diets containing MLP had similar ADG and ADFI. The hot carcass weight was also not different between the two groups. However, pigs fed MLP diets seemed to have lower (*p* < 0.05) backfat thickness while higher (*p* < 0.05) lean percentage. Notably, the hot carcass weight appeared to be lowered by MLP supplementation.

### 3.2. Oxidative Status

As shown in Figure 1, dietary inclusion of MLP did not affect the enzyme activity of serum SOD and MDA. However, compared with CON pigs, MLP pigs appeared to have higher (*p* < 0.05) CAT activity in serum.

### 3.3. Meat Quality

As shown in Table 5, the meat quality of LT muscle in pigs from two treatment groups was determined. Compared with that from CON pigs, the muscle obtained from MLP pigs 24 h postmortem appeared to show a higher L (lightness) value. Moreover, dietary MLP supplementation significantly decreased (*p* < 0.05) the muscle shear force when compared with the CON group, indicating better meat tenderness in MLP pigs. However, the other parameters reflecting meat quality were not different between the two groups.

### 3.4. Muscle Fiber Morphology Traits

Representative characteristics of LT muscle were determined to compare the morphometry traits of myofibers in Tibetan pigs from two groups (Figure 2). The muscle fiber density in MLP pigs was significantly higher (*p* < 0.05) than that in CON pigs. Meanwhile, a smaller (*p* < 0.05) fiber cross-sectional area was observed in the MLP group when compared with the CON group.

### 3.5. Muscle Fiber Type and Related Gene and Protein Expression

To determine the effects of dietary inclusion of MLP on myofiber type transformation, slow-twitch, and fast-twitch fibers were determined using immunofluorescence. No significant difference was observed between the two groups pertaining to the percentage of slow-twitch and fast-twitch fibers (Figure 3). Gene expression results indicated that dietary MLP supplementation significantly up-regulated (*p* < 0.05) the gene expression of myosin heavy chain (MyHC) IIa (Figure 4). PGC1-α is an important regulator in muscle fiber transformation and, thus, was determined. The western blot results showed that, compared with the pigs in the control group, the MLP pigs appeared to have higher (*p* < 0.05) protein expression levels of PGC1-α (Figure 5).

## 4. Discussion

In recent years, indigenous Chinese pork is becoming increasingly popular because of its special flavor and consumers’ different needs [29]. Tibetan pig is a unique Chinese pig breed living in the Qinghai–Tibet plateau region, and its pork is considered to have better taste and flavor [30]. However, the lean percentage of Tibetan pigs is lower than commercial pig breeds such as Yorkshire pigs [9], which may hinder the popularization of Tibetan pork further. To better enlarge the market of Tibetan pork, it is of great importance to find strategies improving the meat quality while controlling the fat contents. The dietary regime has been proven to be effective mays regulating the meat quality of pigs. Given the importance of mulberry bioactive components in regulating obesity and oxidative stress [31], the present study was conducted to study whether dietary supplementation of MLP could impact the lean percentage and meat quality of Tibetan pigs and the potential mechanisms.

Over the years, several studies have evaluated the usage of MLP in pig feeding. However, cross-bred pigs (Duroc × Landrace × Yorkshire, Landrace × Yorkshire × Duroc) were used in these studies [15,19,20]. Given the difference in fat deposition and muscle development patterns [32], it is of great meaning to assess the effects of dietary MLP on growth and muscle development in Tibetan pigs. According to the previous studies, the growth performance of pigs was affected by the supplemental level of MLP, as 4% MLP in the diet was shown to increase the ADG and ADFI of pigs [20], whereas 15% MLP in the diet significantly decreased the ADG of pigs. To maximize the function of bioactive components in MLP and minimize the disadvantageous effect on growth performance, a medium dosage (8%) of MLP was used in our study. Not surprisingly, medium dosage brought neutralized results, as no difference concerning ADG and ADFI in Tibetan pigs was observed following MLP supplementation. Therefore, the effects of MLP on growth performance seem to be dose-dependent.

Although the growth performance was not altered by dietary MLP, the backfat thickness was decreased in Tibetan pigs fed diets containing MLP. The decreased backfat thickness, which is in accordance with a previous study [15], might be attributed to the enhanced lipolysis activity [19]. The increased lean percentage was also observed in MLP-fed pigs in our study, which was not reported in previous studies. The difference in pig breeds could possibly contribute to the inconsistency in lean percentage alteration in response to dietary MLP, as Tibetan pigs can have 20% less lean percentage than commercial pig breeds [12].

With the development of society and the improvement of citizens’ life levels, consumers’ demands for meats are gradually switched from quantity to quality [33]. For the current pig industry, it is of great importance to focus on the improvement of meat quality [34]. Bioactive components in plants, such as polyphenol and flavonoids, which are rich in mulberry leaves, have been proven to play critical roles in regulating meat quality [35,36]. In this study, following dietary MLP supplementation, several parameters reflecting meat quality showed improvement to different extents. For example, the lightness of LT muscle tended to increase following dietary MLP supplementation. According to previous research, supplementation with antioxidants significantly increased the lightness of LT muscle in pigs [37]. Therefore, the improved antioxidant capacity caused by dietary MLP, as evidenced by increased serum catalase activity, might contribute to the alteration of meat color. In a recent study, the improved meat quality in pigs fed chlorogenic acid was attributed to increased antioxidant activity [38]. Hence, the increased antioxidant activity in this study might partially participate in the regulation of meat quality.

Shear force is a fast, accurate, and easy-to-perform technique used to evaluate the tenderness of meat [39]. Herein, the shear force in MLP pigs appeared to be lower than in CON pigs, suggesting improved meat tenderness. It has been reported that the shear force value is negatively correlated with the fiber density while positively correlated with the fiber cross-sectional area [40]. Consistent with this, the H&E staining results in our study showed increased fiber density and decreased fiber cross-sectional area in MLP pigs. Therefore, the alteration of muscle fiber morphology partly contributed to the improvement in meat tenderness. Muscle fibers are usually categorized into four types, typeI, typeIIa, typeIIb, and typeIIx, and the reprogramming of myofiber phenotype changes the contractile and metabolic properties of muscles, thereby impacting the meat quality [41]. Accordingly, gene expression of the four MyHC isoforms was determined, and up-regulated expression of MyHCIIa was observed. TypeIIa myofiber has been reported to be negatively associated with shear force in pig LT muscle [42]. Hence, the improved tenderness of LT muscle might also be related to the change in myofiber types. Apart from the gene expression results, we used an immunofluorescence assay to distinguish fast-twitch (typeII) and slow-twitch fibers (typeI). Similar to commercial pig breeds, most myofibers in Tibetan pigs are fast twitch fibers [43], accounting for more than 85% of the total number. At the transcriptional level, only typeIIa mRNA expression was altered, which seems insufficient to change the percentage of fast-twitch as well as slow-twitch fibers because typeIIb and typeIIx mRNA expression stay unchanged. Chen et al., reported down-regulated MyHCIIb and MyHCIIx mRNA expression in MLP (4%) fed pigs, which is inconsistent with our results. Consistent with our results, in another study, the gene expression of these two MyHC isoforms was not altered when 15% MLP was included in diets. Therefore, the supplemental level of MLP appeared to be a key factor influencing the myofiber type.

PGC-1α, a potent stimulator of mitochondrial biogenesis, plays a key role in maintaining muscle metabolism and controls numerous genes affecting muscle morphology and physiological function [44,45]. PGC-1α is also a pivotal factor regulating muscle fiber type determination [46]. In the present study, dietary MLP stimulated the protein level PGC-1α. In pigs, increased protein expression of PGC-1α has been considered to contribute to better meat quality [37,47]. Collectively, the improved antioxidant activity and altered muscle fiber morphology and type induced by dietary MLP seemed to jointly contribute to the improvement of meat quality in Tibetan pigs. Although myofiber type has been confirmed to be robustly relevant to meat quality [48] and dietary regulation has been revealed to influence the myofiber type of pigs [38,47,49], it is still obscure how to precisely control the transformation of certain myofiber types through nutritional ways, which requires further research to unveil the modulatory mechanisms.

## 5. Conclusions

In conclusion, dietary supplementation of 8% MLP could decrease the backfat thickness, promote lean percentage, increase serum antioxidant activity and improve the meat quality of Tibetan pigs, whereas it had no adverse effects on the growth performance. The improved meat quality, as indicated by improved meat tenderness and meat color, appeared to be the combined action of enhanced antioxidant activity and altered muscle fiber type and muscle fiber morphology. The present study shows feeding strategy is useful for improving the meat quality while decreasing fat deposition in Tibetan pigs and may, thus, further promote the popularity of Tibetan pork.

## Figures and Tables

**Figure 1 animals-12-02743-f001:**
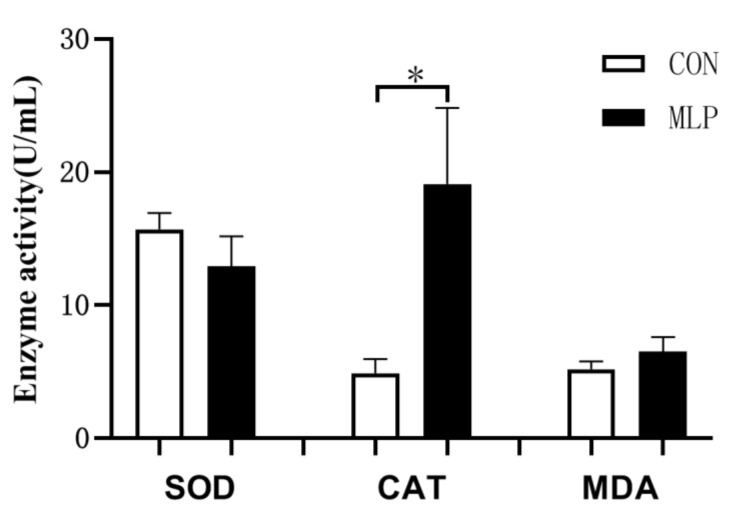
Effect of dietary supplementation with mulberry leaf powder (MLP) on serum antioxidant parameters in Tibetan pigs. Data are expressed as means ± SEM. * indicates a significant difference (*p* < 0.05) between the MLP group and the control (CON) group. N = 9.

**Figure 2 animals-12-02743-f002:**
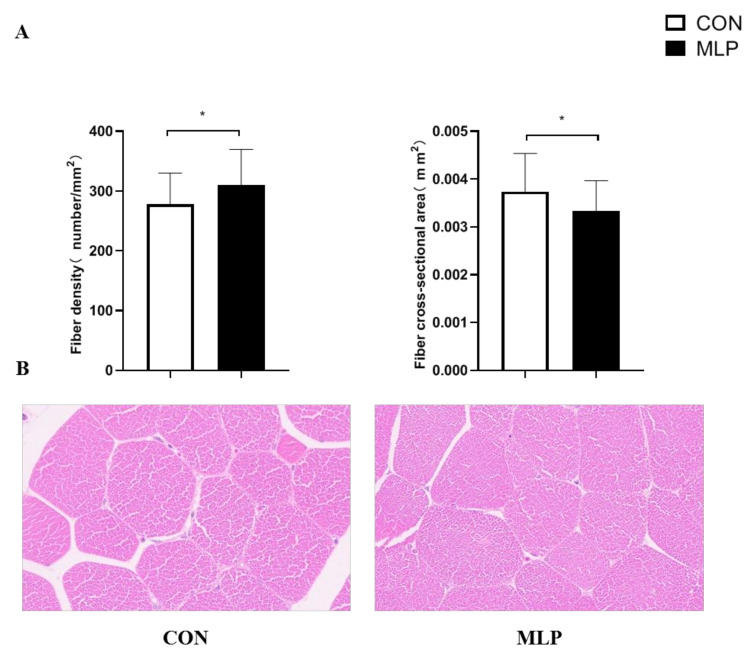
Effect of dietary supplementation with mulberry leaf powder (MLP) on histological characteristics of longissimus thoracis (LT) muscle in Tibetan pigs. (**A**) H&E staining determined the fiber density and fiber cross-sectional area of LT muscle in Tibetan pigs. (**B**) Representative images showing histological characteristics of LT muscle. Data are expressed as means ± SEM. * indicates a significant difference (*p* < 0.05) between the MLP group and the control (CON) group. N = 6.

**Figure 3 animals-12-02743-f003:**
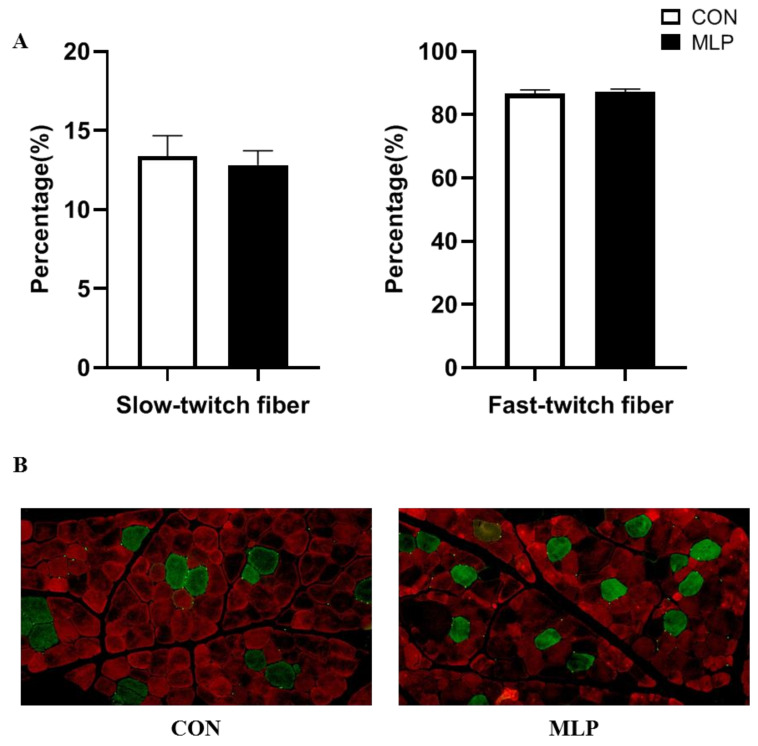
Effect of dietary supplementation with mulberry leaf powder (MLP) on muscle fiber type conversion in longissimus thoracis (LT) muscle of Tibetan pigs. (**A**) Immunofluorescence analyzed the percentage of fast-twitch (red) and slow-twitch (green) fibers in the longissimus thoracis (LT) muscle of Tibetan pigs. (**B**) Representative images showing fiber types in LT muscle. Data are expressed as means ± SEM. N = 6.

**Figure 4 animals-12-02743-f004:**
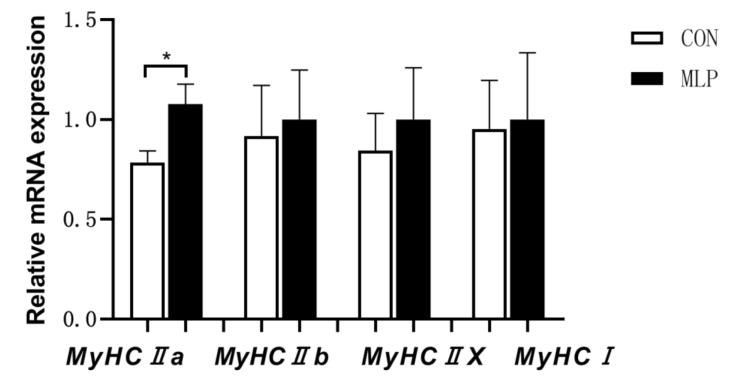
Effect of dietary supplementation with mulberry leaf powder (MLP) on mRNA expression of myosin heavy-chain (MyHC) isoform genes in longissimus thoracis (LT) muscle of Tibetan pigs. Data are expressed as means ± SEM. * indicates a significant difference (*p* < 0.05) between the MLP group and the control (CON) group. N = 6.

**Figure 5 animals-12-02743-f005:**
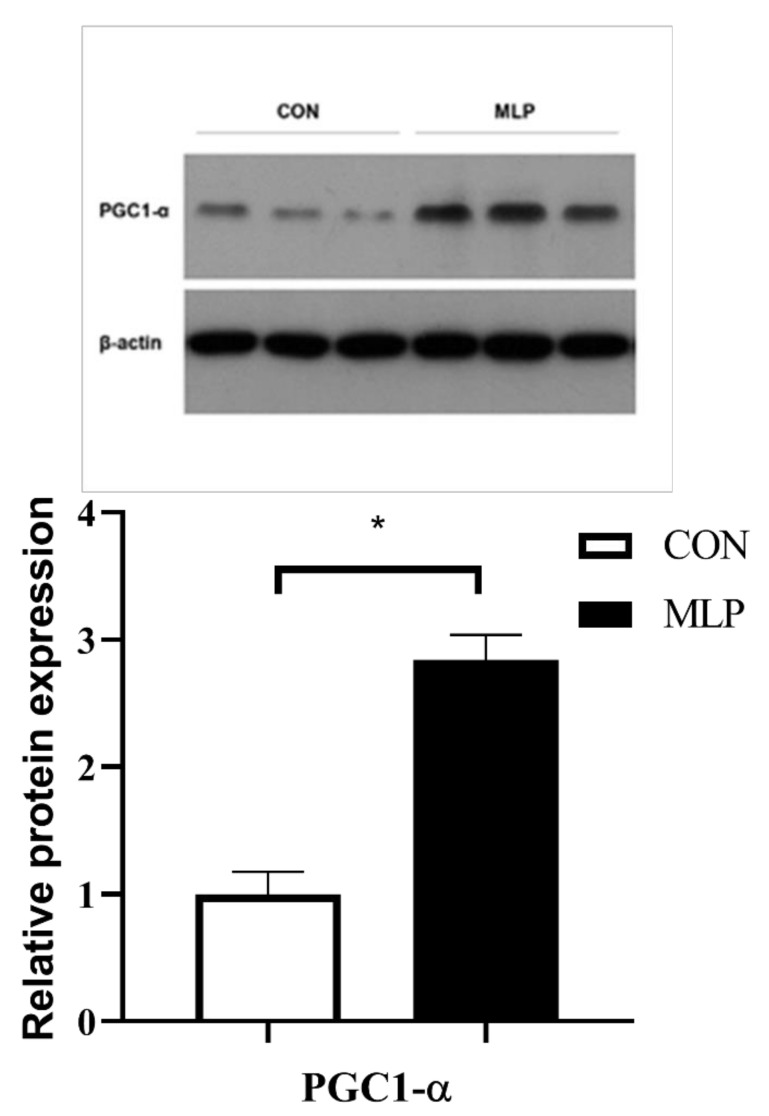
Effect of dietary supplementation with mulberry leaf powder (MLP) on the protein level of peroxisome proliferator-activated receptor-γ coactivator-1α (PGC1-α) in longissimus thoracis (LT) muscle of Tibetan pigs. Data are expressed as means ± SEM. * indicates a significant difference (*p* < 0.05) between the MLP group and the control (CON) group. N = 6.

**Table 1 animals-12-02743-t001:** Nutrient composition of mulberry leave powder (MLP).

Ingredients	Content
Crude protein, %	13.96
Crude fat, %	4.30
Crude fiber, %	6.00
Crude ash, %	8.30
Total calcium, %	1.32
Total phosphorus, %	0.34

**Table 2 animals-12-02743-t002:** Feed ingredients and nutrient composition of the experimental diet (as-fed basis).

Items	Treatment	
	CON	MLP
Ingredients, %		
Corn	62.00	56.50
Soybean meal	25.80	24.50
Rice bran meal	8.20	7.20
Mulberry leaf powder	-	8.00
Wheat middling	1.00	1.00
Soybean oil	1.00	0.80
Vitamin-mineral premix^1^	2.00	2.00
Calculated content		
Digestible energy, Mcal/kg	3.32	3.33
Crude protein	17.61	17.58
Crude fiber	3.16	3.40
Calcium	0.61	0.71
Total phosphorus	0.70	0.69
Standardized ileal digestible amino acids		
Lysine, %	0.76	0.72
Methionine + cysteine, %	0.47	0.43
Threonine, %	0.54	0.51
Tryptophan, %	0.17	0.16

The premix provided the following amounts per kilogram of diet: Fe, 375 mg as iron sulfate; Cu, 12.5 mg as copper sulfate; Mn, 75 mg as manganese sulfate; Zn, 40 mg as zinc oxide; Se, 0.3 mg as sodium selenite; Ca, 5 g as calcium iodate and calcium hydrophosphate; P, 2.2 g as calcium hydrophosphate; Vitamin A, 3200 IU; Vitamin D_3_, 2500 IU; Vitamin E, 60 mg; Vitamin K, 6 mg; Vitamin B_2_, 6 mg; Vitamin B_6_, 5 mg; Vitamin B_12_, 30 μg; Biotin, 0.3 mg; folic acid, 1.0 mg.

**Table 3 animals-12-02743-t003:** Primer sequences for quantitative PCR.

Genes	Primer Sequences (5′ to 3′)	Accession Number	Size, bp
β-actin	FW: CCACGAAACTACCTTCAACTC	XM_003124280.5	131
	RV: TGATCTCCTTCTGCATCCTGT		
MyHCI	FW: AAGGGCTTGAACGAGGAGTAGA	NM_213855.2	114
	RV: TTATTCTGTTCCTCCAAAGGG		
MyHCⅡa	FW: GCTGAGCGAGCTGAAATCC	NM_214136.1	137
	RV: ACTGAGACACCAGAGCTTCT		
MyHCⅡx	FW: AGAAGATCAACTGAGTGAACT	NM_001104951.2	149
	RV: AGAGCTGAGAAACTAACGTG		
MyHCⅡb	FW: ATGAAGAGGAACCACATTA	NM_001123141.1	166
	RV: TTATTGCCTCAGTAGCTTG		

**Table 4 animals-12-02743-t004:** Effects of dietary inclusion of mulberry leaf powder on the growth performance and carcass traits of Tibetan pigs.

Items	Treatment		*p*-Value
	CON	MLP	
Initial BW, kg	34.22 ± 3.03	33.44 ± 3.05	0.87
BW at Day30, kg	41.50 ± 2.60	42.11 ± 2.35	0.87
Final BW, kg	51.28 ± 2.92	50.28 ± 2.63	0.81
ADG, g	309.70 ± 18.34	267.60 ± 10.92	0.12
ADFI, g	1386.00 ± 23.97	1348.00 ± 30.12	0.38
F:G	4.66 ± 0.06	5.05 ± 0.18	0.11
Backfat thickness, cm	29.50 ± 1.57	25.06 ± 1.03	0.04
Hot carcass weight, kg	32.82 ± 0.80	30.18 ± 1.12	0.09
Lean percentage	43.80 ± 0.50	45.76 ± 0.64 *	0.04

* indicates significant difference (*p* < 0.05) between MLP group and control (CON) group. N = 9 for the growth performance results, while N = 6 for the carcass trait results.

**Table 5 animals-12-02743-t005:** Effects of dietary inclusion of mulberry leaf powder on the meat quality of Tibetan pigs.

Items	Treatment		*p*-Value
	CON	MLP	
pH_45min_	6.24 ± 0.09	6.33 ± 0.03	0.32
pH_24h_	5.54 ± 0.19	5.67 ± 0.15	0.60
L* (lightness)_45min_	43.22 ± 1.24	42.21 ± 1.52	0.62
a* (redness)_45min_	20.57 ± 0.73	20.71 ± 0.19	0.86
b* (yellowness)_45min_	2.56 ± 0.27	2.40 ± 0.16	0.63
L* (lightness)_24h_	51.38 ± 1.56	48.08 ± 0.87	0.09
a* (redness)_24h_	22.42 ± 0.86	21.98 ± 0.76	0.70
b* (yellowness)_24h_	7.98 ± 0.75	6.84 ± 0.54	0.24
Drip loss at 24h, %	4.85 ± 0.54	4.69 ± 0.42	0.82
Drip loss at 48h, %	7.87 ± 0.69	7.44 ± 0.65	0.66
Marbling scores	1.47 ± 0.07	1.72 ± 0.35	0.48
Cooking loss	14.40 ± 1.61	16.04 ± 1.44	0.47
Shear force, N	32.72 ± 1.44	27.51 ± 1.65 *	0.04

* indicates significant difference (*p* < 0.05) between MLP group and control (CON) group. N = 6.

## Data Availability

The raw datasets used and analyzed during the current study are available from the corresponding author upon reasonable request.

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
