# Peer review of "The Effects of Dietary Inclusion of Mulberry Leaf Powder on Growth Performance, Carcass Traits and Meat Quality of Tibetan Pigs"

_animals, 2022, doi:10.3390/ani12202743_

Round 1

Reviewer 1 Report

This article introduced a Chinese indigenous pig breed, Tibetan pigsand, and investigated the effect of dietary mulberry leaf powder (MLP) on the meat quality of Tibetan pigs. The authors certified that the MLP improved the meat quality in Tibetan pigs by enhancing antioxidant activity and altering muscle fiber type and morphology.

The idea was good, however, there are still some questions.
Firstly, there are many grammatical errors.

Secondly, in the part about Animals, experimental design, and diets, the body weight records were missing on day 30 of the trial. The diet was in a loose or granular form? Provide a method of mixing the MLP supplement into the diet.

Thirdly, please specify what type of commercial kit you use, Elisa or others.

Fourthly, for Table 5,  the Shear force P<0.05, but there was no significant marker; for the Fig.3 data, the digestible energy is confusing, The statistical graph (Fig.3A) shows that group MLP has less slow-twitch than CON, contrary to Fig.3B?.

The format of the P-value should be uniform throughout the text.

There are many grammatical errors, such as:

L46, It seems that the verb serve does not agree with the subject. Consider changing the verb form.
L48,
It appears that the adverb increasingly is attempting to modify the noun number. Consider replacing the adverb with an adjective.

L59, It seems that month may not agree in number with other words in this phrase.
L60,
The plural verb are does not appear to agree with the singular subject Tibetan pork. Consider changing the verb form for subject-verb agreement.
And so on. Please examine the manuscript carefully and revise it. You can get assistance from a good English writer.

Reviewer 2 Report

1) Line22. Please provide the unit of the weight of pigs, 33.8±1.1 kg?

2) Line76 “effects” and Line154 “inverted fluorescence”, please delete the revision trace.

3) Line116. How could the author slaughter the pigs; did the selected pigs received any aesthesia dose before slaughtering?

4) Line 120. Where and how long the sample has to be better defined. Was this the first 8 cm of Longissimus dorsi collected caudal to the 13/14th ribs? Please define better. Define the location.

5) Line124-137, Please provide more details about measurement of carcass traits and meat quality.

6) Line 158-161. Please provide the product code of CAT, SOD, MDA, and the next BCA kit and others like first antibody.

7) Line123. The author give brief information about the RNA extraction method, how could they collecting RNA, how could they preventing isolated RNA from contamination, the applied annealing temperature and selected genes, the applied method,…. Etc. this section need to described well?

Reviewer 3 Report

Dear Authors.

The paper entitled `The effects of dietary inclusion of mulberry leaf powder on growth performance, carcass traits and meat quality of Tibetan pigs´. It is an interesting work, improving the quality of pork meat by using mulberry leaves, thus expanding the information currently available. The work was written in an understandable and coherent language.

However, the article needs to be clarified in several places to improve its comprehensibility and its scientific quality.

In the "Introduction" section:

-          Line 53: please clarify in the text what are the special living habits of Tibetan pigs.

-          Line 77: remove the underscore for the word “effect”.

-          Lines 78-80: In this study, ....... the effects of MLP, this paragraph should be in the material and methods section, not in the introduction.

In the "materials and methods" section:

2.2 Animal:

 - Line 96 were the 18 pigs randomly selected from a commercial population?

-Please clarify if the 18 pigs included in the experiment were all females or all males?

- The conditions of the production system should be described in the text.

-The experimental design is not clear, because if only 9 pigs were taken per treatment group, the N of the study is not sufficiently representative. Why so few pigs were taken, because it is a breed with low production, although in the introduction it is commented that it is a breed currently very popular in consumption. To clarify and justify why the N of this study is so small, because of the difficulty of raising this breed, because it is an endangered breed, etc. Otherwise, the study would not have a high scientific quality because the population of pigs under study is not representative.

- Also clarify if the nine pigs per group were all raised in the same pen or if they were randomly divided into several pens. The latter would be correct.  

-Specify in the text the procedures of transport, and slaughter, were they stunned with gas, and where were they slaughtered? Specify in the text the regulations governing these procedures.

 - Do the authors have data on the composition of the proximal carcass? If so, I think it would be interesting to include it in the text.

- Please indicate the commercial reference of the diet ingredients in the text.

2.3 Sample collection

- Line 120: Scientifically the nomenclature longissimus dorsi (LD) is not correct, please modify it in line 120 and throughout the manuscript and replace it by longissimus thoracis et lumborum.

-Line 122: how long the samples were frozen?

For the analysis of meat quality, was the muscle cut into fillets? of what thickness? How many fillets were analyzed from each muscle per animal? Clarify in the text how many muscle samples were obtained in total from each animal.

2.4 Measurement

- Indicate in the text the references of each analytical method used in this study: color, pH, shear force and marbling.

 The number of replicates per treatment is not clear in the manuscript. How many replicates were made of each analytic per sample, was it done in duplicate, triplicate?

Also, explain for example the color where it was measured directly on the muscle or on the surface of the fillet.

-Line 134-135: please indicate which scale was used to measure the marbling of the meat (reference).

- Line 154: remove the underscore of the words “inverted fluorescence”.

So my advice is for the authors to clarify the study design.

Regarding the statistical analysis is insufficient. The authors should include all model terms, and all effects. They should clarify the replicates of the study.

-Line 191-192: I think it is too hasty to consider trend P <0. 1 the difference when the N of the study is so small.

- P should be in capitals and italics.

In the title of the tables put a dot after the number and capitalize the word Feed in table 2. Also, unify the number of decimals in all the tables, and put two decimals.

In general, a greater comparison and discussion of the results of this study is lacking.

Round 2

Reviewer 3 Report

Dear Authors.

The paper entitled `The effects of dietary inclusion of mulberry leaf powder on growth performance, carcass traits and meat quality of Tibetan pigs´  fits the scope of the journal well, as you have reliably responded to all comments and improved the manuscript considerably.